Robustness analysis of metabolic predictions in algal microbial communities based on different annotation pipelines

Karimi Elham ekarimi@sb-roscoff.fr 1
Geslain Enora 1 2
Belcour Arnaud 3
Frioux Clémence 4
Aïte Méziane 3
Siegel Anne 3
Corre Erwan 2
Dittami Simon M. simon.dittami@sb-roscoff.fr 1
1 UMR8227, Integrative Biology of Marine Models, Sorbonne Université/CNRS, Station Biologique de Roscoff , Roscoff , France
2 FR2424, Sorbonne Université/CNRS, Station Biologique de Roscoff , Roscoff , France
3 Equipe Dyliss, Univ Rennes, Inria, CNRS, IRISA , Rennes , France
4 Inria, INRAE, CNRS, Univ. Bordeaux , Talence , France
Gillespie Joseph
Electronic publication date: 2021 May 6
Publication date: 2021
Volume: 9
Electronic Location ID: e11344
Received 2020 Oct 8; Accepted 2021 Apr 3
Copyright: ©2021 Karimi et al.
Copyright year: 2021
Copyright holder: Karimi et al.
License: This is an open access article distributed under the terms of the Creative Commons Attribution License, which permits unrestricted use, distribution, reproduction and adaptation in any medium and for any purpose provided that it is properly attributed. For attribution, the original author(s), title, publication source (PeerJ) and either DOI or URL of the article must be cited.
License URL: https://creativecommons.org/licenses/by/4.0/

Keywords: Gene prediction, Functional annotation, Genome-scale metabolic networks, Metabolic complementary analyses, Metabolic exchanges, Holobionts

Funding: CNRS Momentum call and the ANR project IDEALG ANR-10-BTBR-04 Investissements d’Avenir, Biotechnologies-Bioressources This work was supported partially by the CNRS Momentum call and the ANR project IDEALG [ANR-10-BTBR-04] “Investissements d’Avenir, Biotechnologies-Bioressources”. The funders had no role in study design, data collection and analysis, decision to publish, or preparation of the manuscript.

==============================
Animals, plants, and algae rely on symbiotic microorganisms for their development and functioning. Genome sequencing and genomic analyses of these microorganisms provide opportunities to construct metabolic networks and to analyze the metabolism of the symbiotic communities they constitute. Genome-scale metabolic network reconstructions rest on information gained from genome annotation. As there are multiple annotation pipelines available, the question arises to what extent differences in annotation pipelines impact outcomes of these analyses. Here, we compare five commonly used pipelines (Prokka, MaGe, IMG, DFAST, RAST) from predicted annotation features (coding sequences, Enzyme Commission numbers, hypothetical proteins) to the metabolic network-based analysis of symbiotic communities (biochemical reactions, producible compounds, and selection of minimal complementary bacterial communities). While Prokka and IMG produced the most extensive networks, RAST and DFAST networks produced the fewest false positives and the most connected networks with the fewest dead-end metabolites. Our results underline differences between the outputs of the tested pipelines at all examined levels, with small differences in the draft metabolic networks resulting in the selection of different microbial consortia to expand the metabolic capabilities of the algal host. However, the consortia generated yielded similar predicted producible compounds and could therefore be considered functionally interchangeable. This contrast between selected communities and community functions depending on the annotation pipeline needs to be taken into consideration when interpreting the results of metabolic complementarity analyses. In the future, experimental validation of bioinformatic predictions will likely be crucial to both evaluate and refine the pipelines and needs to be coupled with increased efforts to expand and improve annotations in reference databases.

Introduction

Plants, animals, and algae are hosts to a large diversity of microorganisms. The importance of these symbiotic microbes for the development and functioning of their hosts is widely accepted (Amin et al., 2015; Fraune & Bosch, 2010; McFall-Ngai et al., 2013; Philippot et al., 2013). This is also true for brown algal surfaces, which provide an attractive substrate for different bacterial phyla, most importantly the Proteobacteria, Bacteroidetes, Firmicutes, and Actinobacteria. These bacteria are important for the brown algal host e.g., by promoting its growth and by providing defenses against pathogens (KleinJan et al., 2017; Singh & Reddy, 2014). The mutualistic relationships between algae and their microbiota are so tightly interwoven that both parts together are frequently considered a functional entity, the so-called holobiont (Egan et al., 2013).

Recent advances in sequencing technology have generated new opportunities for genome sequencing and genomic analyses of microbial populations (Dittami et al., 2021). A key step in interpreting this wealth of genomic data is their structural and functional annotation, and multiple prokaryotic annotation pipelines have been developed to accomplish this task (Siezen & Van Hijum, 2010). Based on functional annotations, genome-scale metabolic networks can then be generated, providing a formal representation of an organisms’ metabolism (Haggart et al., 2011). Finally, metabolic complementarity can highlight potentially beneficial metabolic interactions between a host and its symbionts (Dittami, Eveillard & Tonon, 2014; Muller et al., 2018), and the degree of metabolic complementarity between two organisms has been suggested to be a direct indicator of beneficial vs. competitive interaction in a system (Burgunter-Delamare et al., 2020; Levy et al., 2015). It can even be used as a criterion for the selection of minimal consortia where the host attains maximal metabolic potential while minimizing exchanges (Frioux et al., 2018; Levy et al., 2015). Although these tools and pipelines provide a promising framework for a comprehensive understanding of the metabolism of organisms, it is currently unknown to what extent outcomes of such analyses are sensitive to gene-associations present in the metabolic networks they rely on, and thus the precision of the prokaryotic annotation pipelines used. For metabarcoding analyses, for instance, the choice of the bioinformatic pipeline has been shown to have a significant impact on some of the biological conclusions that could be drawn (Siegwald et al., 2019).

Here we compared a selection of widespread genome annotation pipelines, examined their impact on the reconstructed genome-scale metabolic networks, and assessed whether the same conclusion about metabolic complementarity could be reached with the different pipelines. Our work was carried out using the filamentous brown alga Ectocarpus subulatus as a model system. Different strains of Ectocarpus have been established as a genomic and genetic model (Cock et al., 2010) and an increasing amount of information is available on its microbiome (Burgunter-Delamare et al., 2020; Dittami et al., 2016; Tapia et al., 2016), including a collection of 72 genomes (Karimi et al., 2019) of bacteria isolated from Ectocarpus as a host (Dittami et al., 2020b).

Materials & Methods

Genome annotation

Eighty-one publicly available genomes were used for this study (Table S1), all of them corresponding to strains that have been isolated from the brown alga Ectocarpus subulatus. The genomes were structurally and functionally annotated using five popular annotation pipelines: the Magnifying Genomes (MaGe) platform (Vallenet et al., 2019), the Rapid Annotations using Subsystems Technology (RAST) toolkit (Aziz et al., 2008), the Integrated Microbial Genomes (IMG) server (Markowitz et al., 2009), the Prokka tool v1.13 (Seemann, 2014) and the DDBJ Fast Annotation and Submission Tool (DFAST) (Tanizawa, Fujisawa & Nakamura, 2017). Each pipeline uses one or more unique databases to predict and search for genes and gene functions. MaGe searches for functional features using UniProtKB/Swiss-Prot, Interpro, FIGFAM, COG, ENZYME, and Diamond as a search tool (Vallenet et al., 2019); RAST predicts gene functions using the SEED database (Aziz et al., 2008); IMG predicts features for genes based on COGs, Pfams, TIGRFAMs, as well as the KEGG and MetaCyc; Prokka is a command-line tool using UniProt, Pfam, and TIGRFAMs; DFAST uses ortholog searches with reciprocal BLAST searches, HMM searches against TIGRFAMs, and COG assignments for functional annotations (Tanizawa, Fujisawa & Nakamura, 2017). All of the pipelines were run with default parameters (Table S2). For Prokka, we also performed tests with more stringent e-value settings (1e−15 vs 1e−6 (default)); as these results were similar to those obtained with default parameters they are included only in the supplementary tables.

Draft metabolic network reconstruction

The draft metabolic network of the host alga Ectocarpus subulatus was already available (Dittami et al., 2020a) and had been reconstructed using Pathway Tools version 23 (Karp et al., 2019) and the AuReMe pipeline (Aite et al., 2018). To be able to use this metabolic network of the host for complementarity analyses, the same version of Pathway Tools was also used to reconstruct the bacterial draft metabolic networks. For each annotated bacterial genome (i.e., 5 pipelines × 81 strains), a draft metabolic network was automatically reconstructed using Mpwt, the PathoLogic multiprocessing wrapper Python package (Belcour et al., 2020) taking into account the protein description, GO terms, and complete EC numbers. The output files created by Pathway Tools were then converted to the SBML format with the Padmet software. During this process the reactions without gene association were removed. Then, the list of reactions as well as network statistics were extracted using the report_network.py script from the Padmet software (Aite et al., 2018). Consequently, for each of the five annotation pipelines, a set of 81 draft metabolic networks was obtained and used for metabolic complementarity analyses. Furthermore, one additional set was produced for each strain by merging the metabolic networks obtained for each annotation pipeline.

Predicting metabolic complementarity

The SBML files were utilized to predict potential metabolic complementarity between the algal host and the bacteria using the MiSCoTo tool (Frioux et al., 2018). Based on a list of compounds present in the culture medium, i.e., Provasoli-enriched seawater as defined by Prigent et al. (2014), MiSCoTo predicts the production of compounds (i.e., the scope) in combined metabolic networks based on a discrete abstraction, which assumes recursively that a compound is producible either if it belongs to the culture medium if it is the product of a reaction in the combined metabolic networks whose substrates are themselves producible, as stated in the network expansion algorithm (Ebenhöh, Handorf & Heinrich, 2004). It thus does not consider the kinetics, or stoichiometry of the reactions, nor the relative abundance of the organism corresponding to the metabolic networks. MisCoTo furthermore allows for the selection of minimal symbiont communities that enable the production of a given set of compounds. In our analyses of the data obtained with the five different pipelines individually plus one analysis merging all genomes from all pipelines, the set of host metabolites that became producible through symbiotic cooperation was computed with miscoto_scopes. This set was then used as a target for community selection with miscoto_mincom using “--soup” as an option, i.e. ignoring the putative cost of exchanges associated with ensuring the producibility of these novel metabolites by the host. In each experiment, the minimal community size ensuring the producibility of these algal compounds was computed, together with the set of symbionts that appears in at least one of the minimal communities (“--union” option). Both the list of producible metabolites and the selected microbial communities were compared for all tested annotation pipelines. The detailed description of the code and data used for the above steps is available on GitHub: https://github.com/ElhamKarimi/Metabolic-predictions_different-Pipelines.

Indirect measures of network quality and manual examination of pipeline-specific reactions reconstructed metabolic networks

To gain insight into the quality of metabolic networks generated based on the different pipelines, dead-end metabolites were computed for each pipeline using Memote (Lieven et al., 2020). Furthermore, sizes of the largest strongly connected components were counted in each of the networks to obtain a measure of its overall connectivity. First, directed metabolite graphs were constructed from the SBML files using the Networkx package (Hagberg, Swart & Chult, 2008). Then, 23 ubiquitous metabolites were removed (water, proton, ATP, ADP, 3-5-ADP, AMP, NADP, NADPH, NAD, NADH, FAD, UDP, GTP, GDP, oxygen molecule, inorganic phosphate, diphosphate, carbon dioxide, ammonia, hydrogen peroxide, coenzyme A, H2-acceptor, H2-donor). Finally, for each graph, strongly connected components were found with the Networkx strongly_connected_components function. Then, for each metabolic network, the ratio between the size of the largest strongly connected component and the total number of metabolites was computed.

In addition to these indirect measures, we randomly chose 100 reactions that were exclusively found in one pipeline (20 per pipeline) for manual curation. Protein sequences of the bacterial genes that led to the prediction of these reactions were aligned with the Swissprot/Uniprot database using blastP and were then classified as high confidence (>30% ID; >70% coverage with a protein with the same metabolic function, functional domains present), low confidence (lower similarity with a protein with the same function, or similarity with a protein with a distinct but similar function), or false (>30% ID; >70% coverage with proteins known to carry out a different function).

Statistical analyses

All statistical analyses were carried out using RStudio (v.1.1.463). Differences in the number of annotation features per pipeline were determined using an ANOVA followed by a Tukey’s HSD (honestly significant difference) test, considering each genome as a replicate. Nonmetric multidimensional scaling (NMDS) was performed on a presence/absence matrix comprising all reactions for each genome/annotation pipeline. The Bray-Curtis index was used as a measure of quantification of compositional dissimilarity between genomes annotated by different pipelines. To verify if the annotation pipeline had an impact on the reaction, an ANOSIM test (number of permutations = 9999) was performed using the Vegan package (Oksanen et al., 2020). UpSet diagrams were generated using the UpSetR package (Conway, Lex & Gehlenborg, 2017). Comparisons of the overall numbers of predicted EC-numbers per category were tested using a Binomial test and a Bonferroni correction for multiple testing.

Results

81 bacterial genomes were annotated using the Prokka, RAST, IMG, MaGe, and DFAST pipelines, and the number of predicted features (Coding Sequences (CDSs), Enzyme Commission (EC) numbers, hypothetical proteins) are shown for each genome and pipeline in Table S1. The number of predicted CDSs across all genomes and pipelines ranged from 2180 to 6755; the number of predicted EC-numbers ranged from 443 to 2106. The number of hypothetical proteins ranged from 56 to 3566. Finally, the average number of predicted metabolic reactions, after automatic metabolic network reconstruction with Pathway Tools, ranged from 1034 to 2052. Figure 1 shows the impact of the tested annotation pipelines on these metrics. At the level of CDS prediction, we observed divergence with MaGe, for instance, predicting more CDSs per genome than all other pipelines. Also, MaGe annotation contained fewer unknown hypothetical proteins.

Figure 1 Boxplot of the number of (A) CDSs, (B) EC-numbers (considers all EC-number), (C) Hypothetical proteins, and (D) Metabolic reactions of all annotated bacterial species through the five pipelines.

The graphs show the deviation of each genome annotated with every one pipeline from the mean across all pipelines in %. The thick horizontal line represents the median of the distribution across all genomes for that pipeline, and circles represent outlier data. Letters above the box-plots indicate statistically significant differences (Tukey’s HSD test). Pipelines share the same letter, the differences between them are not significant (p ≥ 0.05). (See Table S10 for details).

Prokka predicted more EC numbers than the other pipelines, regardless of the e-value cutoff used. The precision of the predictions was similar in Prokka, IMG, MaGe, and DFAST with 85, 88, 84, and 85% of all EC numbers being complete. The only outlier was RAST, which provided only complete EC numbers (Table S3). We did, however, observe significant differences between the pipelines regarding certain categories of EC numbers. For instance, DFAST predicted most oxidoreductases acting on NADH/NADPH, IMG predicted most Ligases forming phosphoric ester bonds, MaGe was the only pipeline to predict translocases (E.C. 7.-.-.-), Prokka predicted most hydrolases acting on phosphorus-nitrogen bonds, and RAST predicted most hydrolases acting on ether bonds. A detailed list of overrepresented EC numbers per annotation pipeline is provided in Table 1.

Table 1 Overrepresented EC numbers in the tested pipelines compared to the average of all pipelines.

	Description	%over	p	
DFAST	1.6 Oxidoreductases acting on NADH or NADPH	52%	<0.001	
	2.1 Transferase transferring one-carbon groups	12%	<0.001	
	2.7 Transferase transferring phosphorus-containing groups	8%	<0.001	
	3.6 Hydrolases acting on acid anhydrides	22%	<0.001	
	6.1 Ligases forming carbon-oxygen bonds	17%	<0.001	
	6.3 Ligases forming carbon-nitrogen bonds	11%	<0.001	
IMG	2.1 Transferase transferring one-carbon groups	7%	0.019	
	3.1 Hydrolases acting on ester bonds	9%	<0.001	
	3.4 Hydrolases acting on peptide bonds	14%	<0.001	
	3.6 Hydrolases acting on acid anhydrides	15%	<0.001	
	6.5 Ligases forming phosphoric ester bonds	68%	<0.001	
MaGe	1.19 Oxidoreductases acting on reduced flavodoxin as donor	5658%	<0.001	
	4.3 Carbon–nitrogen lyases	17%	0.036	
	5.6 Isomerases altering macromolecular conformation	∞	<0.001	
	7.1 Translocases catalysing the translocation of hydrons	∞	<0.001	
	7.2 Translocases catalysing the translocation of inorganic cations and chelates	∞	<0.001	
	7.3 Translocases catalysing the translocation of inorganic anions	∞	<0.001	
	7.5 Translocases catalysing the translocation of carbohydrates and derivatives	∞	<0.001	
	7.6 Translocases catalysing the translocation of other compounds	∞	<0.001	
Prokka	1.1 Oxidoreductases acting on CH-OH group of donors	10%	<0.001	
	1.10 Oxidoreductases acting on diphenols and related substances as donors	56%	<0.001	
	1.12 Oxidoreductases acting on hydrogen as donors	196%	<0.001	
	1.13 Oxygenases	37%	<0.001	
	1.14 Oxidoreductases acting on paired donors, with oxygen	78%	<0.001	
	1.16 Oxidoreductases oxidizing metal ions	56%	<0.001	
	2.4 Glycosyltransferases	21%	<0.001	
	2.8 Transferase transferring sulfur-containing groups	16%	0.004	
	3.1 Hydrolases acting on ester bonds	15%	<0.001	
	3.2 Glycosylases	15%	0.004	
	3.4 Hydrolases acting on peptide bonds	9%	0.025	
	3.6 Hydrolases acting on acid anhydrides	12%	<0.001	
	3.8 Hydrolases acting on halide bonds	86%	<0.001	
	3.9 Hydrolases acting on phosphorus-nitrogen bonds	856%	<0.001	
	4.5 Carbon–halide lyases	∞	<0.001	
RAST	1.2 Oxidoreductases acting on the aldehyde or oxo group of donors	18%	<0.001	
	2.3 Acyltransferases	11%	<0.001	
	2.7 Transferase transferring phosphorus-containing groups	11%	<0.001	
	3.3 Hydrolases acting on ether bonds	72%	<0.001	
	4.1 Lyases Carbon–carbon lyases	15%	<0.001	
	4.2 Carbon–oxygen lyases	12%	<0.001	
	5.1 Isomerases - racemases, epimerases	17%	0.008	
	5.3 Isomerases - intramolecular oxidoreductases	16%	0.004	
	5.4 Isomerases - intramolecular transferases	28%	<0.001	
	6.1 Ligases forming carbon-oxygen bonds	12%	0.010	
	6.3 Ligases forming carbon-nitrogen bonds	10%	0.002	
	6.4 Ligases forming carbon–carbon bonds	41%	<0.001	
Notes.

The “%over” indicates how many more EC numbers were found in a specific pipeline compared to all pipelines; “ ∞” indicates that the EC category was predicted only in this pipeline. p-values correspond to a binomial test after Bonferroni correction.

In terms of predicted metabolic reactions, IMG predicted the highest numbers (4.2% more than average) closely followed by Prokka with default settings (3.1%), while DFAST annotations resulted in the fewest reactions (7.7% fewer than average). Similar patterns were also obtained when running the analyses separately for the different bacterial phyla (Fig. S1), except that EC-number predictions for Actinobacteria by DFAST were lower compared to other pipelines, and CDS prediction for Firmicutes by IMG were slightly higher. Finally, we examined three indirect measures of quality for the network reconstructions: the percentage of dead-ends, i.e., metabolites which can be produced but are not consumed; orphan metabolites, i.e., metabolites that are consumed but not produced in the metabolic networks; and the ratio between the size of the largest strongly connected components and the total number of metabolites, a direct estimate of the “connectedness” of the network. The percentage of dead-end metabolites was similar across all pipelines: 30.35% for Prokka, 29.66% for IMG, 29.17 for DFAST, 29.02% for MaGe, and 28.4% for RAST. The percentage of orphan metabolites was also similar: 30.63% for Prokka, 30.49% for IMG, 29.81% for DFAST, 30.37% for MaGe, and 29.48% for RAST. Lastly, this was also true for the ratio between the size of the largest of strongly connected components and the total number of metabolites: 26% for Prokka, 28% for IMG, 29% for DFAST, 27% for MaGe, and 29% for RAST.

To determine how similar networks based on different annotation pipelines were, the Bray-Curtis dissimilarity, a unidimensional distance measure between two matrices or networks, was calculated for all network comparisons. Non-Metric Multidimensional Scaling (NMDS) was then used to display the resulting distance matrix in the two-dimensional space. In the resulting graph (Fig. 2 and Table S4) the distance between two points represents the dissimilarity of the underlying networks. It shows that the similarity between networks is determined by the bacterial phyla, and less so by the annotation pipeline, but within each phylum, there was neither a clear clustering according to strain nor according to annotation pipeline. ANOSIM analyses were then used to determine if these patterns were statistically significant. They show that both phylum and annotation pipeline had a significant effect on the similarity of the networks (p < 0.0001 in both cases), but confirm that the effect was stronger for phylum (R = 0.93) than for annotation pipeline (R = 0.41).

Figure 2 Non-metric Multi-Dimensional Scaling (NMDS) plot of the metabolic networks of the different bacterial strains and pipelines.

The Bray-Curtis index was used as a dissimilarity measure; ellipses show clusters of strains based on bacterial phyla; grey polygons connect annotations of the same genome performed with different pipelines.

To further explore the differences in the metabolic network reconstructions and to highlight the specificities of each annotation pipeline we aimed to determine the conserved and unique content between the genome annotations from these different pipelines. This was done using the UpSetR package. Besides generating lists of reactions specific to different subsets of pipelines, this package also generates an UpSet diagram (Fig. 3), which displays the size of these sets. For instance, 2535 reactions were predicted in at least one genome with each of the 5 pipelines. Prokka and MaGe annotations resulted in the most pipeline-specific reactions: 390 (325 with more stringent e-value) and 256, respectively. These two tools also shared a high number of pairwise exclusive metabolic reactions, i.e., reactions not found by any other pipeline (307 vs 15–65 for all other pairwise comparisons). The lists of reactions exclusively found in each pipeline as well as lists of the metabolic pathways they belong to are provided in Tables S5 and S6.

Figure 3 Upset chart showing the overlap in predicted biochemical reactions (A) and producible compounds by the algal host and the complete set of 81 bacteria (scope) (B) annotated with each pipeline.

The horizontal bars on the left show the total number of predicted reactions (A) or producible compounds (B) in each pipeline. The vertical histogram on the right shows the number of overlapping reactions (A) or compounds (B). The total number of reactions and producible compounds for each tool is indicated on the left as ‘set size per tool’.

To get an overview of the dominant metabolic functions specific to the networks based on the different pipelines, we examined pipeline-specific metabolic reactions at the level of the pathways they participate in. In total, 180 pathways comprised the reactions specific to Prokka, and among them, 28 were constituted of >50% Prokka-specific reactions, including nitrobenzene degradation (PWY-5637) I, taurine degradation I (PWY-1263), albaflavenone biosynthesis (PWY-5887), and novobiocin biosynthesis (PWY-7287). Similarly, for MaGe the specific reactions could be associated with 127 pathways, and 11 were constituted of >50% specific reactions, including nitroethane degradation (PWY-5355), sulfite oxidation I (PWY-527), and sulfite oxidation IV (PWY-5326). For RAST 99 pathways including 6 pathways >50% complete presented the specific reactions, e.g., acetaldehyde biosynthesis II (PWY-6330) and trypanothione biosynthesis (TRYPANOSYN-PWY). IMG-specific reactions represented 52 pathways with only 4 >50% complete, including tRNA splicing II (PWY-7803) and aldoxime degradation (P345-PWY). Finally, reactions predicted specifically by DFAST corresponded to 42 pathways with only Nitrogen fixation II (PWY-7576) and Fatty acid biosynthesis (PWY-5970) being >50% complete (Table S6).

To evaluate the accuracy of the annotation pipelines, manual curation of 20 randomly selected reactions that were unique to each pipeline was carried out. DFAST has the highest number of high confidence assignations (10) followed by Prokka and MaGe (6), IMG(4), and RAST(1). RAST, on the other hand, did not produce any annotations that could be clearly identified as false, followed by Prokka (1), DFAST (2), IMG (3), and MaGe (3). In Prokka, increasing the stringency of the e-value cutoff did not affect the proportions of high confidence, low confidence or false reactions. In DFAST, although the correct reaction was associated with the gene, we observed discrepancies between the product name of genes and the EC number associated in 3 cases. In MaGe and IMG we found 1 and 3 predicted proteins without any similarity in the database and classified them as low confidence (Table S7).

To assess how these differences in the draft metabolic networks impacted the function of the predicted metabolism of algal–bacterial holobionts, we next examined the list of metabolites that could be produced by the algal metabolic network when combined with the 81 draft bacterial networks for each annotation pipeline (i.e., the predicted producible compounds). As shown in Table 2, based on Prokka, RAST, IMG, MaGe, and DFAST 506, 492, 484, 549, and 504 producible compounds were identified, respectively (see Table S8 for a list of compounds). 448 (i.e., between 81.6% and 92.6% of the producible compounds) were shared among all tools, but MaGe, RAST, and Prokka also had 34, 11, and 10 exclusive compounds (Fig. 3B). Furthermore, 12 additional compounds became producible by merging the networks of all strains and annotation pipelines.

Table 2 Selection of minimal bacterial communities based on annotation pipelines.

Strains	Phylogeny	Prokka	IMG	MaGe	DFAST	RAST	Merged	
Curtobacteriumsp. 8I-2	Actinobacteria(1); Actinobacteria(1); Actinomycetales(1); Microbacteriaceae(1); Curtobacterium(1);					✓	✓	
Microbacteriumsp. 8M	Actinobacteria(1); Actinobacteria(1); Actinomycetales(1); Microbacteriaceae(1); Microbacterium(1);					✓	✓	
Plantibactersp. T3	Actinobacteria(1); Actinobacteria(1); Actinomycetales(1); Microbacteriaceae(1); Plantibacter(1);					✓	✓	
Pseudoclavibactersp. 8L	Actinobacteria(1); Actinobacteria(1); Actinomycetales(1); Microbacteriaceae(1); Pseudoclavibacter(1);	✓			✓	✓	✓	
Arthrobactersp. 8AJ	Actinobacteria(1); Actinobacteria(1); Actinomycetales(1); Micrococcaceae(1); Arthrobacter(1);					✓	✓	
Arthrobactersp. 9V	Actinobacteria(1); Actinobacteria(1); Actinomycetales(1); Micrococcaceae(1); Arthrobacter(1);		✓				✓	
Citricoccussp. K5	Actinobacteria(1); Actinobacteria(1); Actinomycetales(1); Micrococcaceae(1); Citricoccus(1);			✓		✓	✓	
Micrococcussp. 116	Actinobacteria(1); Actinobacteria(1); Actinomycetales(1); Micrococcaceae(1); Micrococcus(1);	✓				✓	✓	
Micrococcussp. 11B	Actinobacteria(1); Actinobacteria(1); Actinomycetales(1); Micrococcaceae(1); Micrococcus(1);					✓	✓	
Micrococcussp. 80W	Actinobacteria(1); Actinobacteria(1); Actinomycetales(1); Micrococcaceae(1); Micrococcus(1);					✓	✓	
Aeromicrobiumsp. 9AM	Actinobacteria(1); Actinobacteria(1); Actinomycetales(1); Nocardioidaceae(1); Aeromicrobium(1);	✓				✓	✓	
Nocardioidessp. AX2bis	Actinobacteria(1); Actinobacteria(1); Actinomycetales(1); Nocardioidaceae(1); Nocardioides(1);		✓			✓	✓	
Imperialibactersp. 89	Bacteroidetes(1);Cytophagia(1); Cytophagales(1); Flammeovirgaceae(1); Imperialibacter(1);			✓	✓		✓	
Imperialibactersp. EC-SDR9	Bacteroidetes(1);Cytophagia(1); Cytophagales(1); Flammeovirgaceae(1); Imperialibacter(1);			✓	✓			
Imperialibactersp. SDR9	Bacteroidetes(1); Cytophagia(1); Cytophagales(1); Flammeovirgaceae(1); Imperialibacter(1);			✓	✓		✓	
Flavobacteriumsp. 9R	Bacteroidetes(1); Flavobacteriia(1); Flavobacteriales(1); Flavobacteriaceae(1); Flavobacterium(1);	✓	✓			✓	✓	
Frigoribacteriumsp. 9N	Bacteroidetes(1); Flavobacteriia(1); Flavobacteriales(1); Flavobacteriaceae(1); Flavobacterium(1);	✓				✓	✓	
Maribactersp. 151	Bacteroidetes(1); Flavobacteriia(1); Flavobacteriales(1); Flavobacteriaceae(1); Maribacter(1);	✓	✓	✓	✓	✓	✓	
Sphingobacteriumsp. 8BC	Bacteroidetes(1); Sphingobacteriia(1); Sphingobacteriales(1); Sphingobacteriaceae(1); Sphingobacterium(1);					✓	✓	
Bacillussp. 348	Firmicutes(1); Bacilli(1); Bacillales(1); Bacillaceae 1(1); Bacillus(1);					✓		
Bacillussp. 349Y	Firmicutes(1); Bacilli(1); Bacillales(1); Bacillaceae 1(1); Bacillus(1);				✓	✓		
Bacillussp. 71	Firmicutes(1); Bacilli(1); Bacillales(1); Bacillaceae 1(1); Bacillus(1);					✓		
Bacillussp. 9J	Firmicutes(1); Bacilli(1); Bacillales(1); Bacillaceae 1(1); Bacillus(1);	✓				✓		
Exiguobacteriumsp. 8A	Firmicutes(1); Bacilli(1); Bacillales(1); Bacillales_Incertae Sedis XII(1); Exiguobacterium(1);					✓	✓	
Exiguobacteriumsp. 8H	Firmicutes(1); Bacilli(1); Bacillales(1); Bacillales_Incertae Sedis XII(1); Exiguobacterium(1);					✓	✓	
Exiguobacteriumsp. 9Y	Firmicutes(1); Bacilli(1); Bacillales(1); Bacillales_Incertae Sedis XII(1); Exiguobacterium(1);					✓	✓	
Flavobacteriumsp. 9AF	Firmicutes(1); Bacilli(1); Bacillales(1); Bacillales_Incertae Sedis XII(1); Exiguobacterium(1);	✓	✓	✓		✓	✓	
Staphylococcussp. 8AQ	Firmicutes(1); Bacilli(1); Bacillales(1); Staphylococcaceae(1); Staphylococcus(1);		✓	✓		✓	✓	
Brevundimonassp. G8	Proteobacteria(1); Alphaproteobacteria(1); Caulobacterales(1); Caulobacteraceae(1); Brevundimonas(1);					✓		
Boseasp. 125	Proteobacteria(1); Alphaproteobacteria(1); Rhizobiales(1); Bradyrhizobiaceae(1); Bosea(1);				✓	✓		
Bosea sp. 127	Proteobacteria(1); Alphaproteobacteria(1); Rhizobiales(1); Bradyrhizobiaceae(1); Bosea(1);				✓	✓		
Boseasp. 21B	Proteobacteria(1); Alphaproteobacteria(1); Rhizobiales(1); Bradyrhizobiaceae(1); Bosea(1);				✓	✓		
Boseasp. 29B	Proteobacteria(1); Alphaproteobacteria(1); Rhizobiales(1); Bradyrhizobiaceae(1); Bosea(1);				✓	✓		
Boseasp. 46	Proteobacteria(1); Alphaproteobacteria(1); Rhizobiales(1); Bradyrhizobiaceae(1); Bosea(1);				✓	✓		
Boseasp. 62	Proteobacteria(1); Alphaproteobacteria(1); Rhizobiales(1); Bradyrhizobiaceae(1); Bosea(1);				✓	✓		
Boseasp. 7B	Proteobacteria(1); Alphaproteobacteria(1); Rhizobiales(1); Bradyrhizobiaceae(1); Bosea(1);				✓	✓		
Hoefleasp. HK425	Proteobacteria(1); Alphaproteobacteria(1); Rhizobiales(1); Phyllobacteriaceae(1); Hoeflea(1);					✓		
Rhizobiumsp. SD404	Proteobacteria(1); Alphaproteobacteria(1); Rhizobiales(1); Rhizobiaceae(0.96); Rhizobium(0.92);	✓	✓	✓			✓	
Roseovariussp. EC-HK134	Proteobacteria(1); Alphaproteobacteria(1); Rhodobacterales(1); Rhodobacteraceae(1); Roseovarius(1);			✓	✓	✓	✓	
Roseovariussp. SD190	Proteobacteria(1); Alphaproteobacteria(1); Rhodobacterales(1); Rhodobacteraceae(1); Roseovarius(1);			✓	✓	✓	✓	
Erythrobactersp. HK427	Proteobacteria(1); Alphaproteobacteria(1); Sphingomonadales(1); Erythrobacteraceae(1); Erythrobacter(1);	✓				✓		
Novosphingobiumsp. 9U	Proteobacteria(1); Alphaproteobacteria(1); Sphingomonadales(1); Sphingomonadaceae(1); Novosphingobium(1);	✓				✓		
Sphingomonassp. 8AM	Proteobacteria(1); Alphaproteobacteria(1); Sphingomonadales(1); Sphingomonadaceae(1); Sphingomonas(1);					✓		
Sphingomonassp. AX6	Proteobacteria(1); Alphaproteobacteria(1); Sphingomonadales(1); Sphingomonadaceae(1); Sphingomonas(1);					✓		
Sphingomonassp. EC-HK361	Proteobacteria(1); Alphaproteobacteria(1); Sphingomonadales(1); Sphingomonadaceae(1); Sphingomonas(1);					✓		
Sphingomonassp. EC-SD391	Proteobacteria(1); Alphaproteobacteria(1); Sphingomonadales(1); Sphingomonadaceae(1); Sphingomonas(1);			✓		✓	✓	
Sphingomonassp. T1	Proteobacteria(1); Alphaproteobacteria(1); Sphingomonadales(1); Sphingomonadaceae(1); Sphingomonas(1);					✓	✓	
Sphingorhabdussp. 109	Proteobacteria(1); Alphaproteobacteria(1); Sphingomonadales(1); Sphingomonadaceae(1); Sphingorhabdus(1);					✓	✓	
Burkholderiasp. 8Y	Proteobacteria(1); Betaproteobacteria(1); Burkholderiales(1); Burkholderiaceae(1); Burkholderia(1);			✓			✓	
Burkholderiales bacterium 8X	Proteobacteria(1); Betaproteobacteria(1); Burkholderiales(1); Comamonadaceae(1); Variovorax(1);			✓		✓	✓	
Massiliasp. 9I	Proteobacteria(1); Betaproteobacteria(1); Burkholderiales(1); Oxalobacteraceae(1); Massilia(1);					✓		
Aeromonassp. 8C	Proteobacteria(1); Gammaproteobacteria(1); Aeromonadales(1); Aeromonadaceae(1); Aeromonas(1);	✓	✓			✓	✓	
Aeromonassp. 9A	Proteobacteria(1); Gammaproteobacteria(1); Aeromonadales(1); Aeromonadaceae(1); Aeromonas(1);	✓	✓		✓		✓	
Alteromonassp. 154	Proteobacteria(1); Gammaproteobacteria(1); Aeromonadales(1); Aeromonadaceae(1); Aeromonas(1);					✓		
Alteromonassp. 38	Proteobacteria(1); Gammaproteobacteria(1); Alteromonadales(1); Alteromonadaceae(1); Alteromonas(1);					✓		
Marinobactersp. HK377	Proteobacteria(1); Gammaproteobacteria(1); Alteromonadales(1); Alteromonadaceae(1); Marinobacter(1);			✓	✓	✓	✓	
Marinobactersp. N1	Proteobacteria(1); Gammaproteobacteria(1); Alteromonadales(1); Alteromonadaceae(1); Marinobacter(1);				✓	✓	✓	
Pantoeasp. 111	Proteobacteria(1); Gammaproteobacteria(1); Enterobacteriales(1); Enterobacteriaceae(1);Pantoea(1);	✓				✓		
Enterobacterales bacterium 8AC	Proteobacteria(1); Gammaproteobacteria(1); Enterobacteriales(1); Enterobacteriaceae(1); Yersinia(0.68);		✓		✓	✓	✓	
Halomonassp. 113	Proteobacteria(1); Gammaproteobacteria(1); Oceanospirillales(1); Halomonadaceae(1); Halomonas(1);		✓	✓	✓		✓	
Halomonassp. 153	Proteobacteria(1); Gammaproteobacteria(1); Oceanospirillales(1); Halomonadaceae(1); Halomonas(1);		✓	✓	✓		✓	
Halomonassp. 156	Proteobacteria(1); Gammaproteobacteria(1); Oceanospirillales(1); Halomonadaceae(1); Halomonas(1);		✓	✓	✓		✓	
Halomonassp. 59	Proteobacteria(1); Gammaproteobacteria(1); Oceanospirillales(1); Halomonadaceae(1); Halomonas(1);		✓	✓	✓		✓	
Halomonassp. 98	Proteobacteria(1); Gammaproteobacteria(1); Oceanospirillales(1); Halomonadaceae(1); Halomonas(1);		✓	✓	✓		✓	
Halomonassp. I3	Proteobacteria(1); Gammaproteobacteria(1); Oceanospirillales(1); Halomonadaceae(1); Halomonas(1);		✓		✓			
Acinetobactersp. 8BE	Proteobacteria(1); Gammaproteobacteria(1); Pseudomonadales(1); Moraxellaceae(1); Acinetobacter(1);		✓			✓		
Acinetobactersp. 8I-beige	Proteobacteria(1); Gammaproteobacteria(1); Pseudomonadales(1); Moraxellaceae(1); Acinetobacter(1);					✓	✓	
Enhydrobactersp. AX1	Proteobacteria(1); Gammaproteobacteria(1); Pseudomonadales(1); Moraxellaceae(1); Enhydrobacter(0.85);					✓		
Moraxellaceae bacterium 17A	Proteobacteria(1); Gammaproteobacteria(1); Pseudomonadales(1); Moraxellaceae(1); Enhydrobacter(0.91);				✓	✓		
Enhydrobactersp. 8BJ	Proteobacteria(1); Gammaproteobacteria(1); Pseudomonadales(1); Moraxellaceae(1); Enhydrobacter(1);			✓		✓		
Pseudomonassp. 8AS	Proteobacteria(1); Gammaproteobacteria(1); Pseudomonadales(1); Pseudomonadaceae(1); Pseudomonas(1);		✓			✓	✓	
Pseudomonassp. 8BK	Proteobacteria(1); Gammaproteobacteria(1); Pseudomonadales(1); Pseudomonadaceae(1); Pseudomonas(1);					✓		
Pseudomonassp. 8O	Proteobacteria(1); Gammaproteobacteria(1); Pseudomonadales(1); Pseudomonadaceae(1); Pseudomonas(1);	✓				✓		
Pseudomonassp. 8Z	Proteobacteria(1); Gammaproteobacteria(1); Pseudomonadales(1); Pseudomonadaceae(1); Pseudomonas(1);				✓	✓	✓	
Pseudomonassp. 9Ag	Proteobacteria(1); Gammaproteobacteria(1); Pseudomonadales(1); Pseudomonadaceae(1); Pseudomonas(1);						✓	
Number of compounds producible by the community;	506	484	549	492	504	590	
Size of the minimal solution:	10	7	13	10	9	14	
Total number of bacteria found in at least one solution (union):	15	17	21	28	63	46	
Notes.

Numbers in parentheses denote the proportion of bootstrap replicates supporting this taxonomic assignation.

Next, we computed how many strains were needed to enable the production of the same metabolic compounds as the entire bacterial communities for each pipeline (Table 2). As expected, the annotation pipeline affected the number of strains involved in a minimal community, which increased with the number of metabolites that could be produced by the entire bacterial community. The number of selected strains ranged from seven strains needed to produce the 484 metabolites producible by the metabolic networks reconstructed with the IMG pipeline to 13 strains needed to produce the 549 metabolites producible by the MaGe metabolic networks. Fourteen strains were required to produce the full 590 compounds producible by the combination of all networks from all pipelines. We also examined which bacteria were involved in these minimal communities, taking into consideration the union of all minimal communities proposed by MisCoTo. Here, the annotation pipeline had a strong effect on these communities, with strains being selected from sub-sets of bacteria ranging from 15 (Prokka) to 61 (RAST) strains (Table 2). The number of bacteria in each of the sub-sets was not related to the number of producible compounds nor the size of the minimal communities. Fifteen strains were selected in only one of the five pipelines, 14 of which exclusively in the RAST-based community, and one, Pseudomonas 9AG, was included in minimal communities only when they were calculated from merged data from all annotation pipelines. Most strains were jointly selected by two or more pipelines, but even Prokka and IMG, for which minimal communities were based on the smallest subset of bacteria (15 and 17 selected strains, respectively) had only six strains in common. However, we also found commonalities between the selected communities: all communities comprised at least one representative of the Actinomycetales (Actinobacteria), Bacillales (Firmicutes), Rhizobiales (Alphaproteobacteria), Sphingomonadales (Alphaproteobacteria), and Pseudomonadales (Gammaproteobacteria). Lastly, one strain, Maribacter strain 151 (Flavobacteriaceae, Bacteroidetes), was consistently selected in all data sets regardless of the annotation tool (Table 2).

Finally, given the high variability between the selected communities, we tested how specific these communities were to the dataset used to select them. To this means, we selected the strains comprised in the minimal microbial communities predicted based on the Prokka annotations, i.e., 15 bacteria, and examined the metabolites predicted to be produced by this community together with the host when applied to the datasets based on the other pipelines. Briefly, the DFAST, IMG, MaGe, and RAST metabolic networks with the Prokka community predicted similar scopes as the larger communities specifically selected for the dataset: 477 compounds vs 492 for RAST, 469 vs 484 for IMG, 488 vs 549 for MaGe, and 448 vs 504 producible compounds for DFAST, respectively (Table S9). This suggests that differences in the community size and composition between the pipelines can be explained mainly by the need to complement each community with bacteria enabling the production of less than 10% of the metabolites.

Discussion & Conclusions

In this study, we sought to assess the impact of selected annotation pipelines on draft metabolic network reconstructions and downstream analyses of metabolic complementarity. Our results highlight significant differences between the output of the five tested standard pipelines at all examined levels, from the prediction of coding sequences to the selection of microbial communities, especially for EC annotations and hypothetical proteins, with levels of variability similar to those previously reported by Griesemer et al. (2018). Overall, the number of reactions predicted in the final network mirrored the number of EC numbers predicted, with Prokka and IMG yielding both the highest number of EC annotations and reactions. This underlines the importance of this type of annotation for metabolic reconstructions with Pathway Tools - a link that is not surprising as EC numbers are directly referenced in MetaCyc (Caspi et al., 2017), the database used by Pathway Tools for draft metabolic network reconstruction. Each complete EC number can therefore be translated directly into one or several corresponding metabolic reactions.

Our analyses also highlight a number of reactions that were predicted specifically for some pipelines, but our analysis of the corresponding pathways did not clearly show any pipelines to favor the annotation of specific biological processes over others. Furthermore, the fact that we observed little differences in the relative performance of the pipelines when comparing them separately for different bacterial phyla, suggests that there is no strong phylum-specificity of the tested pipelines. Our manual examination of a subset of pipeline-specific reactions showed that the reliability of these pipeline-specific predictions is generally low, with a tendency for those pipelines that predict the fewest pipeline-specific reactions (DFAST, RAST) to produce the highest proportion of high-confidence prediction, as well as the most connected networks in terms of dead-end metabolites. The overall high proportion of dead-end metabolites can be explained by the facts that (i) during the reconstruction process we discarded reactions without genetic support (gene association), and that (ii) metabolic networks underwent no or very little curation. Furthermore, the genomes analyzed here belong to non-model organisms, and hence their metabolism is more difficult to reconstruct than in well-established models.

Given these differences, a key question is how reliable functional analyses based on these draft metabolic networks are, and how much their results change according to the annotation pipeline employed. Here our results may seem contradictory at first: on one hand, variability at the level of producible metabolites was low; on the other hand, during the selection of microbial communities based on metabolic complementarities, even these small differences in the draft metabolic networks resulted in largely different consortia. These small differences, whether they are erroneous or missing annotations, accumulate across the 81 annotated genomes and are particularly likely to impact the overall set of producible metabolites and hence community selection, as any possible cooperation with an added value to the host is selected. Indeed, although small, differences in the producible metabolites require complementing the bacterial community with strains that specifically enable the production of these metabolites. Overall, however, the low number of pipeline-specific metabolites indicated that all of the selected consortia were functionally similar. Furthermore, we have shown that a given minimal bacterial community (in our case one generated with Prokka data), also yields similar producible metabolites regardless of the annotation pipeline used for the generation of the metabolic networks. This means that the different selected bacterial consortia are likely able to fulfill the same or similar metabolic roles in our metabolic model. This phenomenon has previously also been described in natural alga-associated bacterial consortia (Burke et al., 2011) and was explained by the competitive lottery theory (Sale, 1979): several, but not all species/strains can occupy a given niche, and among them, random processes govern which species prevails (lottery model). In our case, such “random” processes could be generated by the ‘noise’ in the genome annotations and the specificities of each pipeline. Consequently, the niche would correspond to the provision of specific metabolic functions. In line with this analogy, the composition of the bacterial communities in our analyses was not entirely random: although only one strain was present in all communities, each community also contained at least one member of several major taxonomic groups. These groups likely possess specific metabolic capacities absent from other groups making their presence indispensable in all selected consortia. These requirements were detected and met by MiSCoTo regardless of the annotation pipeline.

Regarding the use of metabolic complementarity as a criterion to select microbial communities, this implies that, as also confirmed experimentally by Burgunter-Delamare et al. (2020), metabolic complementarity can be used to select microbial functions important for a symbiotic community. However, we need to expect high variability in the composition of the selected communities if several strains contain similar metabolic capacities. Just like in nature, there may simply not be one ideal solution that clearly excludes all others. In this sense, our data does not provide any evidence that one or another pipeline is more suitable for metabolic network reconstruction—we can only state that for our dataset Prokka and IMG, on average, produced more EC numbers and larger draft metabolic networks with Pathway Tools, while DFAST and RAST are likely to produce fewer false-positive reactions and dead-end metabolites, on average. Furthermore, MaGe produced the largest global scope for the community. One approach to avoid biases introduced by annotation pipelines is to merge results from different pipelines thus maximizing the number of annotations (Kalkatawi, Alam & Bajic, 2015). Our data obtained for the merged metabolic networks of all pipelines suggest that this approach would also further extend our metabolic networks and the global scope of the community. However, this approach also comes with a risk, as every additional annotation pipeline may introduce additional errors (Poptsova & Gogarten, 2010), especially if the pipeline is not regularly updated (Salzberg, 2019). Based on our manual curation of pipeline-specific reactions we may even consider the opposite i.e., basing metabolic complementarity analysis exclusively on reactions that have been predicted independently by two or more pipelines. This will reduce the scope of the networks but likely also result in a reduction of noise and false-positive associations. An additional option may be to increase the stringency of the pipeline settings. Here moderate modification of the e-value in Prokka had little effect on the final results, but further adjustments are possible and their efficiency would likely need to be adapted for each dataset.

Another application of metabolic complementarity using different annotation pipeline could also be in the context of metagenome analysis, on Metagenome Assembled Genomes (MAGs) of uncultured bacteria. Belcour et al. (2020) have shown that reconstructed metabolic networks from MAGs are similar to the reference genomes. Also, the stability of the producible metabolites and selected minimal symbionts was demonstrated for degraded genomes with 2% of genes randomly removed. It, therefore, seems probable that the overall stability observed in our study in terms of the predicted metabolic contributions of the symbionts to the algal metabolism regardless of the annotation pipeline will also hold true for metagenomic communities.

In the long run, continued and extensive experimental validation of bioinformatic predictions in both a culture and a metagenomic context will be key to evaluate and refine the pipelines (Poptsova & Gogarten, 2010) and needs to be coupled with increased efforts to expand and improve annotations in reference databases (Carr & Borenstein, 2014). In the meantime, one approach to overcome the variability brought by the annotation pipelines is to carefully curate the resulting metabolic networks. This is an indispensable step in obtaining high-quality metabolic networks as stated by Thiele & Palsson (2010). However, this step is costly as it usually requires human expertise and thorough literature exploration. As more and more genomes and metagenomes are available, there is a need for curation-free and reliable metabolic networks to surmount this bottleneck. Non-curated metabolic networks can be informative but are likely to contain false-positive functions brought by the annotation, these functions deserve to be examined closely when used for selecting communities of interest.

Supplemental Information

Supplemental Information 1 Supplementary tables

The list of genomes used in the study and their properties (S1), the parameters used for genome annotation (S2), the EC numbers (S3) and reactions (S4) predicted in each genome/annotation pipeline, the pipeline-specific reactions (S5) and pathways (S6), the results of the manual curation for 100 reactions (S7), the metabolites producible by the cooperation of algal network and bacterial networks with each of the pipelines (S8, S9), and the results of the detailed outcomes of the ANOVA test corresponding to Fig. 1 (S10).

Click here for additional data file.

Supplemental Information 2 Violin plot of metabolic reactions, EC-numbers, CDS, and hypothetical proteins for each tool separated by bacterial phylum

The graphs show the deviation of each genome annotated with everyone pipeline from the mean across all pipelines in %.

Click here for additional data file.

We would like to thank Alireza Asvadi for his help writing a python script for merging tables of all strains across pipelines, and the reviewers of this paper for their constructive remarks. We are grateful to the Institut Français de Bioinformatique (ANR-11-INBS-0013) and BioGenouest Roscoff Bioinformatics platform ABiMS (http://abims.sb-roscoff.fr/) for providing computing resources.

Additional Information and Declarations

Competing Interests

Author Contributions

Data Availability

The authors declare there are no competing interests.

Elham Karimi conceived and designed the experiments, performed the experiments, analyzed the data, prepared figures and/or tables, authored or reviewed drafts of the paper, and approved the final draft.

Enora Geslain performed the experiments, analyzed the data, prepared figures and/or tables, and approved the final draft.

Arnaud Belcour and Méziane Aïte analyzed the data, prepared figures and/or tables, developed software, and approved the final draft.

Clémence Frioux conceived and designed the experiments, prepared figures and/or tables, authored or reviewed drafts of the paper, developed software, and approved the final draft.

Anne Siegel and Erwan Corre conceived and designed the experiments, authored or reviewed drafts of the paper, and approved the final draft.

Simon M. Dittami conceived and designed the experiments, analyzed the data, prepared figures and/or tables, authored or reviewed drafts of the paper, and approved the final draft.

The following information was supplied regarding data availability:

The genomes are publicly available in the European Nucleotide Archive (ENA/EMBL):

PRJEB31339 and PRJEB34356. The accession numbers are available in Table S1.

The detailed description of the code and data used for analysing is available at Github: https://github.com/ElhamKarimi/Metabolic-predictions_different-Pipelines.

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
