# Peer review of "Robustness analysis of metabolic predictions in algal microbial communities based on different annotation pipelines"

_PeerJ, doi:10.7717/peerj.11344_

## Round 0.1 · original submission · Major Revisions

Dear Dr. Karimi and colleagues:

Thanks for submitting your manuscript to PeerJ. I have now received three independent reviews of your work, and as you will see, the reviewers raised some major concerns about the research (and the manuscript). Despite this, these reviewers are optimistic about your work and the potential impact it will have on research on bioinformatics approaches to analyzing algal microbial communities. Thus, I encourage you to revise your manuscript, accordingly, taking into account all of the concerns raised by both reviewers.

The overall major criticism across the three reviews pertains to adequately comparing different annotation pipelines. Your revision should strive to be more robust with these explanations and comparisons. Your manuscript also seems to fall short in effectively conveying the broader applicability to the community.

There are many minor problems pointed out by the reviewers, and you will need to address all of these and expect a thorough review of your revised manuscript by these same reviewers.

I agree with the concerns of the reviewers, and thus feel that their suggestions should be adequately addressed before moving forward. Therefore, I am recommending that you revise your manuscript, accordingly, taking into account all of the issues raised by the reviewers.

I look forward to seeing your revision, and thanks again for submitting your work to PeerJ.

Good luck with your revision,

-joe

Reviewer 1 ·

Basic reporting

No comment.

Experimental design

No comment.

Validity of the findings

No comment.

Additional comments

The authors present a comparative analysis about the effect of the choice of gene annotation methods on metabolic reconstructions and on the inference of minimal bacterial communities. The paper is well written and provides the necessary sources and materials.
The topic of this analysis is very important. Indeed, we know that metabolic reconstruction is affected by the identification of enzymatic functions from the genome. But this effect is rarely measured. As shown here, its effect on analyzes performed from metabolic reconstructions can be also very significant. But unfortunately the article does not go very beyond this observation...

Here are my major comments and propositions in order of importance
- As mentioned in the discussion, this analysis does not give us any idea of ​​the sensitivity and specificity of the reaction lists inferred by each analysis. An efficient way to measure their accuracy would have been to compare the result of these analyzes with a PGDB already cleaned manually. Unfortunately, none of the PGDBs in the list seem to have been manually curated by another team. Maybe, it would have been interesting to compare the result of these analyzes with a PGDB already manually cured such as E. coli even if it is not part of this bacterial community. Even without doing this comparative analysis which may seem a little outside the scope of the paper, it would have been interesting to have an idea of ​​the quality of the networks reconstructed by indirect measures, such as the number of connected components in the reconstructed network (without the ubiquitous metabolites) or the number of holes in the pathways identified by the pathway tools.

- It would have been interesting to focus on the prediction quality of EC numbers. It does not seem to me that the authors indicate if they only take into account full EC numbers. It can also be a measure of quality. Also, are certain classes of EC numbers (up to the first 2 or 3 digits) over or under represented by certain pipelines?

- The Pathologic tool of the pathway-tools infers a first list of reactions from the enzymatic annotations, then deduces a list of possible metabolic pathways in the organism. The reactions returned by Pathologic contain this first list of reactions but also the other reactions composing the inferred metabolic pathways, commonly called "pathway holes". The authors do not specify it in the text but I imagine that they took all the reactions, i.e. those associated with a gene annotation and those corresponding to pathway holes. However, we know that these pathway holes can be numerous and that there are many false positives in these inferred metabolic pathways. It would therefore have been very interesting to compare the number of inferred metabolic reactions obtained only by the enzymatic annotations.

- The Miscoto algorithm selects minimal communities. But I imagine that like any result of optimizations, there could be several communities of minimum size, which could change the conclusions made from Table 1. This point is not discussed in the paper.

Minor comments
p8 l126. Authors mentioned 6 experiments but they mentioned above only 5 different analyses.
p8 l136. The github link provided by the authors does not contain "the detailed description of the code and data used...". This link claims that we can find the different files in the cluster but we don't have access to the cluster.

Reviewer 2 ·

Basic reporting

The manuscript by Karimi et al examined the impact on metabolic prediction when different genome annotation software was used. The manuscript is well-written with proper literature reference, sufficient background info. Tables and figures look good too.

Overall, this work by itself has a lot of value in that with the emerging number of new genome annotation tools, how the choice of annotation software impacts the downstream analysis such as metabolic network construction needs careful evaluation so that researchers could be well-informed and choose the proper tools for various research purposes.

Experimental design

This manuscript raised a very good research question and provided some observations.
It is understandable that the authors started with 81 genomes from isolated strains. However, questions like how this evaluation would perform on microbes which have not been isolated and only have incomplete/ near complete genome info, because this is very common in a lot of microbial communities, where most of the microbes could not be cultured. Adding discussions on such topic would greatly enhance the generality and application of the current findings.
In my opinion, it would also greatly strengthen the results if the author could compare the predicted metabolites with the actual measured metabolites in this microbial community, either from other papers’ results or measured in a synthetic community with the 81 isolated microbes and algae. Otherwise, one would not know which predicted metabolites are false positive, and the discussion can only limit to comparing the numbers of different metabolites from pipelines.
Also, it would be nice if the author could share the parameters used for each annotation pipeline for reproducibility.

Validity of the findings

In my opinion, how/whether this knowledge from the manuscript could be translated to other microbial communities is not well-stated. How the presented results could add any insight to the existing knowledge in the field is not clear to me.
Some questions to consider: Now that you see different performance of the annotation pipelines, what is your hypothesis on what may have caused it? Any further work to test the hypothesis? What do you suggest other researchers when they want to do metabolic prediction in a similar situation? Etc…

Additional comments

- Of the 81 strains, how many percent do they cover the whole algal microbial community (e.g. based on culture-independent sequencing method)? Are these 81 strains representative?

- When you do metabolic network reconstruction, how do you define the input (e.g. culture medium)? Did you take into consideration of the abundance info when building the network? As it is very likely these 81 strains do not exist in equal abundance in the community. Any inter-species interactions considered in the modeling?

- Figure 1: any statistical difference among the pipelines?

- Figure 2: It will also be interesting to see the ANOSIM stats for phylum.

Reviewer 3 ·

Basic reporting

Please see general comments below.

Experimental design

Please see general comments below.

Validity of the findings

Please see general comments below.

Additional comments

In the manuscript titled “Robustness analysis of metabolic predictions in algal microbial communities based on different annotation pipelines” Karimi et. al. present a study that compares different genome annotation pipelines and their resulting impact on draft genome scale metabolic reconstructions. The aim of this study is relevant as the number of available microbial genomes is expanding rapidly and the quality of the resulting annotation impacts downstream analyses.

Overall the study design is appropriate and results valid. However, the authors could enhance the utility of the results if there was a more concerted effort to understand the source of the differences between the annotation pipelines.

Comment 1:
The primary weakness of the manuscript is the lack of analysis as to why the different pipelines result in different metrics. Without this assessment, a reader cannot deduce the strengths and weaknesses of each of the pipelines. As a result, the only conclusion from the study is that there are differences between the pipelines, which is largely self-evident since they leverage different databases and search algorithms. While a comprehensive assessment of the differences is out of scope for such a study, a cursory investigation into some of the differences would enhance the manuscript. Some suggested areas to explore:

- The annotation pipelines were run with default settings (Line: 99). Are the homology search cutoffs significantly different in the default settings for the different pipelines? Often homology search is able to assign the biochemistry of the reaction (the first 3 digits of the EC) but the substrate specificity of the 4th digit of the EC is less certain. If the homolog cutoff for one of the pipelines is lower it will call more EC numbers resulting in more reactions, but they will be low confidence. Running a more stringent version of the homology search might address some of the discrepancies in reaction count.

- The authors look at the unique pathways in the draft metabolic reconstructions (Line 183). Perhaps a manual annotation of a few of these pathways might elucidate the source of the different results in the different pipelines?

Comment 2:
The numbers for figure 3 are perplexing. The authors state the Prokka reconstruction had 2535 reactions shared with all the other tools and 390 that were unique to Prokka for a total of 2925 reactions. The Supplementary Table states there are over 133,000 reactions in the 81 microbe consortium. On initial look, this is an exceedingly high amount of redundancy considering the broad taxonomic distributions of the representative microbes. Since it is consistent between all the pipelines, this may be an issue with metabolic reconstruction tool. Since the follow-on analysis includes metabolite production capability, issues with the recon tool will be projected into this analysis and affect the authors’ conclusions about that analysis. This should be addressed by the authors.

Comment 3:
The initial analysis of different metrics between the tools is interesting.However, the statistical analysis needs more context. What is the goal of that analysis? How does it help understand the differences between the annotation pipelines? The Bray-Curtis Index is an interesting mechanism to create a uni-dimensional dissimilarity value from a genome annotation comparison. Unfortunately it isn’t described in the text. Additionally, it isn’t clear how NMDS contributes to an increased understanding of the differences between the different tools. While this analysis encompasses all of Figure 2, there are only 4 lines of text describing the value of this analysis.

Comment 4:
UpSet is the name of a tool, not the name of a known analytical/visualization method. It would be helpful, for almost all of the results section, to develop topic sentences that outline the gap/desired result of the analysis. Then follow up with the tool/method that would facilitate that analysis. An example (Line 173):

“To further explore the differences in the metabolic network reconstructions and to highlight the
specificity of each annotation pipeline we aimed to determine the conserved and unique content between the genome annotations from these different pipelines. An UpSet diagram enables this comparison by…”

As written, the reader doesn’t know why the authors are choosing a specific analytical tool or method nor what the intended outcome of that analysis should reveal and it’s connection with the overall study aims.

Comment 5:
The paragraph starting on line 183 is difficult to understand. It would be helped by use of a topic sentence. It is clear the authors are trying to summarize a very large, interacting set of results, which is difficult. However, the paragraph as written doesn’t clearly communicate the desired results (unique metabolic pathways elucidated in each of the pipelines).

Comment 6:
Line 151: This line is currently written presenting an average and range. Since it comes right after a comment about using the different annotation tools the reader may think the different tools generated this wide range of results. In fact, I believe the authors are saying the different species in the consortium have different genome sizes and they are reporting that distribution. Since this is expected, I would suggest not reporting an average but instead focus on the range of genome sizes in the community.

Comment 7:
Overall, the Discussion and Conclusions are well written. In particular, it was important to point out that a larger number of EC numbers doesn’t necessarily result in a more accurate network; it may in fact result in false positives when tested for metabolic capabilities. The final comment on manual annotation is important but untouched in this work. Similar to the primary weakness, it would enhance the manuscript if there were a few small examples where manual annotation resolved discrepancies between tools.

---

## Round 0.2 · Minor Revisions

Dear Dr. Karimi and colleagues:

Thanks for revising your manuscript. The reviewers are very satisfied with your revision (as am I). Great! However, there are a few minor issues to address. Please address these ASAP so we may move towards acceptance of your work.

Best,

-joe

Reviewer 1 ·

Basic reporting

No new comment.

Experimental design

No new comment.

Validity of the findings

No new comment.

Additional comments

Thanks to the authors for addressing most of my comments. The manuscript is now clearer and richer. I have still one major comment and few minor comments.

Major comments:
- The authors evaluated the quality of the reconstructions by counting the number of orphan and dead end metabolites. They claim that "The lower these values the more connected a metabolic network is". I think that this statement is not always true. We can imagine a network with a lot of dead end (outputs of the network) and orphans (inputs) that contains a single connected component. On the contrary, a network with less dead ends and orphans can be divided into several connected components. I think that the number of connected components (and optimally, the distribution of their sizes and the size of the biggest component) should complete these indices. I think that these connected components should be computed by removing ubiquitous metabolites, such as ATP, water, etc... Furthermore, it's clear that the number of connected components will strongly affect the results of the Miscoto algorithm.



Minor comments:
l 111 : had -> had been (or has been ?)
l 129 : conversion to SBML already mentioned in the previous paragraph
l 190 : In terms of EC numbers, Prokka predicted more than the other pipelines -> I'm not sure that this sentence is grammatically correct.
Fig 2 : What do represent the grey polygones ? If the authors don't comment them, they sould remove them to make the figure clearer.
l 342 : With the new EC number comparative analysis, this statement is not completely true anymore.
l 355 : to reconstructed -> to reconstruct

Reviewer 2 ·

Basic reporting

NA

Experimental design

NA

Validity of the findings

NA

Additional comments

The revised manuscript addresses my comments and questions well and I think it is now ready to be published at Peer J.

Reviewer 3 ·

Basic reporting

See General Comments below.

Experimental design

See General Comments below.

Validity of the findings

See General Comments below.

Additional comments

The changes to the manuscript made by the authors addressed most of my comments. I feel the revision is a much more approachable version of the manuscript and more effectively communicated the results.

There is some proofreading required, but it is minor.

The manual curation of select reactions was a welcome addition. I was hoping for a more clear root-cause for the differences in the pipelines but it seems that it is elusive (no fault of the authors!).

The only thing I would have liked to have seen would been an adjustment of the pipeline parameters to a more stringent version. It is possible the pipelines converge on a largely consistent "core" reconstruction for reactions with high homology but the low homology content is highly affected by the underlying databases used by the different pipelines. In general, the e-value cutoff of 10e-5 or 10e-6 is very forgiving (in my experience of performing manual reconstructions) and will lead to many ambiguous reactions being added. The authors stated in their response that they "deliberately chosen to take the perspective of a standard user" but I would argue the standard user who came across this manuscript would greatly appreciate finding a study that performed some perturbation of the standard parameters.

This also ties into the final comment in the discussion about manual curation and the cost-benefit trade-off of time and reconstruction accuracy. It is very helpful to have an automated process to establish the core reconstruction and flags the ambiguous content for manual curation. I feel like this study is the perfect venue to explore the stringency parameters and the impact on the core reconstruction.

The study is still sound, but I do feel this is a missed opportunity that would increase the scientific contribution of the work.

---

## Round 0.3 · accepted · Accept

Dear Dr. Karimi and colleagues:

Thanks for revising your manuscript based on the concerns raised by the reviewers. I now believe that your manuscript is suitable for publication. Congratulations! I look forward to seeing this work in print, and I anticipate it being an important resource. Thanks again for choosing PeerJ to publish such important work.

Best,

-joe

---

## Author Rebuttal · Round 0.3

## Reviewer 1 (Anonymous)

Thanks to the authors for addressing most of my comments. The manuscript is now clearer and richer. I have still one major comment and few minor comments.

Major comments-
The authors evaluated the quality of the reconstructions by counting the number of orphan and dead end metabolites. They claim that "The lower these values the more connected a metabolic network is". I think that this statement is not always true. We can imagine a network with a lot of dead end (outputs of the network) and orphans (inputs) that contains a single connected component. On the contrary, a network with less dead ends and orphans can be divided into several connected components. I think that the number of connected components (and optimally, the distribution of their sizes and the size of the biggest component) should complete these indices. I think that these connected components should be computed by removing ubiquitous metabolites, such as ATP, water, etc... Furthermore, it's clear that the number of connected components will strongly affect the results of the Miscoto algorithm.

Thank you for this justified remark. Using the Networkx package, we have calculated the size of the largest connected components (after removal of ubiquitous compounds) for all networks. But even after removing ubiquitous components, we have found that 80-90% of metabolites (see Figure 1 below) belong to the largest connected component and thus found this metric to be little informative. In the manuscript, we now instead present the proportion of the largest strongly connected components, which is calculated from directed graphs. The results globally correspond to the data obtained for the dead-end metabolites – the higher the size of the largest strongly connected components, the lower the number of dead-end metabolites (see Figure 2). These results are now described in the manuscript.

[Figure]

**Figure 1.** Ratio between the size of the largest connected components and the total number of metabolites in the metabolic networks generated based on the tested pipelines across all examined genomes. Letters above the box-plots indicate statistically (in)significant differences made by an ANOVA test. Pipelines share the same letter, the differences between them are not significant ($p \geq 0.05$).

[Figure]

**Figure 2.** Ratio between the size of the largest strongly connected components and the total number of metabolites in the metabolic networks generated based on the tested pipelines across all examined genomes. Letters above the box-plots indicate statistically (in)significant differences made by an ANOVA test. Pipelines share the same letter, the differences between them are not significant ($p >= 0.05$).

Minor comments:

l 111 : had -> had been (or has been ?)

Corrected.

l 129 : conversion to SBML already mentioned in the previous paragraph

Deleted in this paragraph.

l 190 : In terms of EC numbers, Prokka predicted more than the other pipelines -> I'm not sure that this sentence is grammatically correct.

Corrected to "Prokka predicted more EC numbers than the other pipelines".

Fig 2 : What do represent the grey polygones ? If the authors don't comment them, they should remove them to make the figure clearer.
Grey polygons connect annotations of the same genome performed with different pipelines. We now added this information to the figure legend.

l 342 : With the new EC number comparative analysis, this statement is not completely true anymore.
Yes, but curation was minimal considering that we looked at 100 reactions in over 300 networks, i.e. less than one reaction per network. We now state: "metabolic networks underwent no or very little curation"

l 355 : to reconstructed -> to reconstruct
Corrected.

## Reviewer 3 (Anonymous)

The changes to the manuscript made by the authors addressed most of my comments. I feel the revision is a much more approachable version of the manuscript and more effectively communicated the results.

There is some proofreading required, but it is minor.

The manual curation of select reactions was a welcome addition. I was hoping for a more clear root-cause for the differences in the pipelines but it seems that it is elusive (no fault of the authors!).

The only thing I would have liked to have seen would been an adjustment of the pipeline parameters to a more stringent version. It is possible the pipelines converge on a largely consistent "core" reconstruction for reactions with high homology but the low homology content is highly affected by the underlying databases used by the different pipelines. In general, the e-value cutoff of 10e-5 or 10e-6 is very forgiving (in my experience of performing manual reconstructions) and will lead to many ambiguous reactions being added. The authors stated in their response that they "deliberately chosen to take the perspective of a standard user" but I would argue the standard user who came across this manuscript would greatly appreciate finding a study that performed some perturbation of the standard parameters.

This also ties into the final comment in the discussion about manual curation and the cost-benefit trade-off of time and reconstruction accuracy. It is very helpful to have an automated process to establish the core reconstruction and flags the ambiguous content for manual curation. I feel like this study is the perfect venue to explore the stringency parameters and the impact on the core reconstruction.

The study is still sound, but I do feel this is a missed opportunity that would increase the scientific contribution of the work.

Thank you for this comment. We agree that improvements to the final networks can probably be made by optimizing the parameter settings, but on the other hand exhaustively and systematically exploring the effect of such changes even in one pipeline could fill an entire study. Please also note that not all pipelines (RAST, MAGE, IMG) allow users to control cutoffs and for those that do, we would need to submit the same dataset to the server multiple times. However, we fully understand your curiosity regarding this point and performed additional analyses to at least touch on it in our manuscript. Notably, we now reannotated all genomes with Prokka 1.13 because it was the fastest of all used pipelines, runs locally, and easily allows for adjustments of parameter settings. For this second round of annotations, we used an e-value cutoff of 1e-15 instead of 1e-6(default) and then compared the results. Not surprisingly, with increasing stringency, we observed a decrease in the number of predicted EC numbers (average 1290 vs 1406) and the number of reactions (1527 vs 1645), and an increase in the number of "hypothetical" proteins (1933 vs 1370). We also examined the effect this had on the Prokka-specific reactions and found this number to decrease from 390 to 325. Among the 20 Prokka-specific reactions that were manually curated three were removed by increasing the threshold (3.2.1.158-RXN, RXN-13750, i.e. RXN-15364), one that had been identified as high confidence, and two that had been identified as low confidence. The overall

proportion of high/low confidence or false reactions remained essentially the same as illustrated below:

|  | e-6 (default) | e-15 |
| --- | --- | --- |
| High confidence | 6 (30%) | 5 (29%) |
| Low confidence | 13 (65%) | 11 (65%) |
| False | 1 (05%) | 1 (06%) |

This result shows that, while parameter settings affect the results, it is not that straight forward to improve them, although there are many more parameters and more levels of stringency to test.

The most important problem for optimizing parameters in our data set is that we are dealing with a multitude of different genomes from different phyla. Thus, the optimal settings are likely to vary. For instance, when annotating a new strain of *E. coli*, it may be beneficial to use a very stringent set of parameters, while, when annotating a novel family or order, less stringent parameters may be necessary. The default parameters were chosen by software/pipeline developers to work for most genomes, and as we deviate from them, we might improve predictions for one genome while obtaining poorer results for others. In the context of the prediction of metabolic consortia, however, varying parameter setting from one genome to the next may introduce new biases.

We have now added the aforementioned results to supplementary tables S1, S5, and S7(gbk files for e-value 1e-15 can be found in the given github link), and briefly mentioned these analyses both in the methods part and in the discussion. We hope you can accept our reasoning as to why we do not wish to make this an integral part of the manuscript.